# Peripherally Restricted CB1 Receptor Inverse Agonist JD5037 Treatment Exacerbates Liver Injury in MDR2-Deficient Mice

**DOI:** 10.3390/cells13131101

**Published:** 2024-06-25

**Authors:** Jenny Chen, Fengyuan Li, Jiyeon Lee, Md Manirujjaman, Lihua Zhang, Zhao-Hui Song, Craig McClain, Wenke Feng

**Affiliations:** 1Department of Medicine, University of Louisville School of Medicine, Louisville, KY 40202, USA; 2Department of Pharmacology and Toxicology, University of Louisville School of Medicine, Louisville, KY 40202, USA; 3Department of Structural and Cellular Biology, Tulane University School of Medicine, New Orleans, LA 70112, USA

**Keywords:** liver fibrosis, Cannabinoids

## Abstract

Previous research highlighted the involvement of the cannabinoid CB1 receptor in regulating the physiology of hepatocytes and hepatic stellate cells. The inhibition of the CB1 receptor via peripherally restricted CB1 receptor inverse agonist JD5037 has shown promise in inhibiting liver fibrosis in mice treated with CCl4. However, its efficacy in phospholipid transporter-deficiency-induced liver fibrosis remains uncertain. In this study, we investigated the effectiveness of JD5037 in *Mdr2^−/−^* mice. *Mdr2 (Abcb4)* is a mouse ortholog of the human MDR3 (ABCB4) gene encoding for the canalicular phospholipid transporter. Genetic disruption of the *Mdr2* gene in mice causes a complete absence of phosphatidylcholine from bile, leading to liver injury and fibrosis. *Mdr2^−/−^* mice develop spontaneous fibrosis during growth. JD5037 was orally administered to the mice for four weeks starting at eight weeks of age. Liver fibrosis, bile acid levels, inflammation, and injury were assessed. Additionally, JD5037 was administered to three-week-old mice to evaluate its preventive effects on fibrosis development. Our findings corroborate previous observations regarding global CB1 receptor inverse agonists. Four weeks of JD5037 treatment in eight-week-old *Mdr2^−/−^* mice with established fibrosis led to reduced body weight gains. However, contrary to expectations, JD5037 significantly exacerbated liver injury, evidenced by elevated serum ALT and ALP levels and exacerbated liver histology. Notably, JD5037-treated *Mdr2^−/−^* mice exhibited significantly heightened serum bile acid levels. Furthermore, JD5037 treatment intensified liver fibrosis, increased fibrogenic gene expression, stimulated ductular reaction, and upregulated hepatic proinflammatory cytokines. Importantly, JD5037 failed to prevent liver fibrosis formation in three-week-old *Mdr2^−/−^* mice. In summary, our study reveals the exacerbating effect of JD5037 on liver fibrosis in genetically MDR2-deficient mice. These findings underscore the need for caution in the use of peripherally restricted CB1R inverse agonists for liver fibrosis treatment, particularly in cases of dysfunctional hepatic phospholipid transporter.

## 1. Introduction

Liver fibrosis is a chronic liver disease characterized by the excessive production and deposition of extracellular matrix proteins, resulting in liver damage and dysfunction [1]. It represents a significant global health burden and currently lacks effective treatment options. There is an urgent need for the development of novel therapeutic targets. Recent advancements have uncovered potential breakthroughs in fibrosis reversal. One such discovery involves the endocannabinoid system (ECS), which plays a critical role in regulating inflammation and fibrogenesis through various components, including cannabinoid CB1 and CB2 receptors, endogenous ligands, and associated enzymes [2,3].

CB1 and CB2 receptors are widely distributed throughout the body, including the liver, where they participate in diverse physiological and pathological processes. While CB1 receptors (CB1Rs) are predominantly expressed in the central nervous system (CNS) and involved in regulating appetite, pain, and mood, they are also present in peripheral tissues, such as the liver, where they modulate lipid metabolism and insulin resistance. CB2 receptors (CB2Rs) are primarily found in immune and hematopoietic cells, with lower levels detected in the liver during certain pathological conditions. CB2Rs have been implicated in anti-inflammation processes and fibrogenesis [4,5].

In healthy liver tissue, endocannabinoid receptors have minimal expression, but studies have revealed upregulation of CB1Rs in hepatic myofibroblasts, vascular endothelial cells, and other nonparenchymal cells, like inflammatory cells, cholangiocytes, and hepatic stellate cells in various liver diseases, including alcohol-associated liver disease, metabolic dysfunction-associated steatotic liver disease (MASLD), hepatic fibrogenesis, and hepatic cirrhosis, among others [6].

Prior research has shown that broad CB1R antagonists like rimonabant effectively reduce body weight in experimental animals and patients with obesity. Genetic and pharmacological inactivation of CB1Rs was also demonstrated to suppress matrix remodeling and the fibrogenic response associated with acute or chronic liver injury, indicating a potential novel antifibrotic role for such drugs [5]. However, due to an increased risk of adverse psychiatric events, such as anxiety, depression, and suicidal ideation, rimonabant was withdrawn from the market [7]. Consequently, the focus has shifted to peripherally restricted CB1R inverse agonists like JD5037 [8], which cannot penetrate the blood–brain barrier and can eliminate the associated CNS risks.

Previous studies have demonstrated that JD5037 reduces body weight gain and attenuates liver fibrosis in mice treated with CCl_4_. CCl_4_ is a toxin that induces hepatocellular damage, inflammation, the activation of hepatic stellate cells, and the subsequent collagen deposition and fibrosis [9]. However, it remains unclear whether the effects of JD5037 on liver fibrosis vary depending on the underlying etiology of liver injury. *Mdr2^−/−^* mice represent a genetic model characterized by impaired biliary phospholipid secretion, leading to cholestasis and progressive liver fibrosis [10].

In this study, our objective was to evaluate the effects of JD5037 in *Mdr2^−/−^* mice. Interestingly, our results revealed that JD5037 significantly exacerbated liver fibrosis in these mice. These findings shed light on the potential challenges associated with the therapeutic application of peripheral CB1 receptor inverse agonist in liver fibrosis treatment, emphasizing the necessity for further investigation into the underlying mechanisms of JD5037’s effect on liver fibrosis.

## 2. Material and Methods

*Animal model*. Mating pairs of *Mdr2^−/−^* mice were purchased from Jackson Laboratory (Bar Harbor, ME). The mice were bred in house to produce animals for the experiment. Eight-week-old *Mdr2^−/−^* male mice were divided into two groups: one group of mice was treated with JD5037 (dissolved in saline, Cayman Chem., Ann Arbor, MI, USA), at a dose of 3 mg/kg of body weight every other day for 4 weeks by oral gavage, while the control group was treated with saline. In an additional experiment, 3-week-old *Mdr2^−/−^* male mice were treated with JD5037 for 5 weeks. The mice were maintained at 22 °C with a 12-h:12-h light/dark cycle and had free access to normal chow diet and water. At the end of the experiment, the mouse tissues and blood were collected and analyzed. All animal protocols were approved by the Institutional Animal Care and Use Committee of the University of Louisville.

*Biochemical assays*. Plasma ALT and ALP levels were measured according to the standard procedure. Briefly, blood samples were collected in EDTA-treated tubes; the samples were kept at room temperature for 30–60 min and then centrifuged at 2000× *g* for 30 min at 4 °C. The supernatant was collected in a new tube. ALT and ALP were measured using a commercially available kit according to the manufacturer’s protocol [11].

*Histological analysis.* Liver sections from paraffin-embedded tissues were prepared at 5 μm thickness. Liver pathology was examined by hematoxylin and eosin staining (H&E). Hepatic fibrosis was assessed by Sirius red staining using standard procedures as described previously [12]. The images were taken via light microscopy at 100× magnifications. 

*Real-time polymerase chain reaction (real-time PCR).* Total RNA was isolated using Trizol according to the manufacturer’s protocol. Total RNA was used for reverse transcription with a cDNA cycle kit (Invitrogen, Carlsbad, CA, USA). Primers are listed in Table 1. 18S was used as an internal control. Real-time qPCR was performed by using SYBR green reaction mixture in an ABI 7300 fast real-time PCR system (Applied Biosystems). The relative gene expression was determined by the ^ΔΔ^CT method. 

*Bile acid (BA) measurements*. Total BA concentrations in liver extracts and serum were determined by Diazyme Total Bile Acids (TBA) Assay according to the manufacturer’s instruction (Diazyme Lab., Poway, CA, USA).

*Statistical analysis.* Statistical analyses were performed using GraphPad Prism, version 9 (GraphPad Software Inc., La Jolla, CA, USA). Results are expressed as means ± SEM. Statistical comparisons were made using one-way ANOVA with Tukey’s post hoc test, or Student *t* test, where appropriate. Differences were considered significant at *p* ≤ 0.05. Significance is noted as * *p* ≤  0.05, ** *p* ≤  0.01, *** *p* ≤  0.001 among groups.

## 3. Results

*JD5037 treatment exacerbated liver injury in Mdr2^−/−^ mice.* To evaluate the effects of JD5037 on liver injury, we utilized 8-week-old *Mdr2^−/−^* mice, which spontaneously developed liver fibrosis. The mice were subjected to JD5037 treatment via oral gavage every other day for 4 weeks, while control mice were administered saline (Figure 1A). In accordance with a previous study [9], JD5037 administration significantly reduced body weight (Figure 1B). However, to our surprise, JD5037 markedly increased liver enzyme ALT and ALP levels, the serological markers of liver injury (Figure 1C,D). Hematoxylin and eosin staining of liver sections showed that *Mdr2^−/−^* mice at the age of 12 weeks experienced significant cell death and liver architecture disruption. JD5037 administration increased hepatocellular damage (green arrows), inflammatory cell infiltration (yellow arrows), and periductal fibrosis (white arrows) (Figure 1E). In addition, hepatic CB1R protein was marginally upregulated by JD5037 treatment, while CB2R protein levels remained unchanged (Figure 1F). These results clearly demonstrated that oral administration of JD5037 exacerbated liver injury in *Mdr2^−/−^* mice.

*JD5037 treatment increased BA levels*. Since *Mdr2^−/−^* mice spontaneously developed cholestasis, we further measured the levels of BA. As shown in Figure 2A,B, JD5037 treatment led to a 2.5-fold increase in circulating BA levels, but it did not affect liver BA levels. However, JD5037 had no effect on the mRNA levels of the liver enzymes involved in BA synthesis (Figure 2C) and transporters (Figure 2D).

*JD5037 treatment exacerbated liver fibrosis*. Eight-week-old mice developed spontaneous fibrosis, shown as some focal points, in the liver. Intriguingly, a 4-week treatment with JD5037 dramatically increased evenly distributed liver fibrosis along the liver lobule, as indicated by Sirius red staining of liver sections (Figure 3A). Moreover, JD5037 treatment markedly elevated liver hydroxyproline levels (Figure 3B), which serve as the marker of collagen content. The mRNA levels of key fibrosis-related factors, including *Col1*, *Timp1*, *Tgfb*, and *aSMA*, were significantly increased by JD5037 treatment (Figure 3C). Additionally, JD5037 treatment significantly increased the expression of CK19, a marker of liver ductular reactions, suggesting bile duct remodeling and cholangiocyte activation. Furthermore, JD5037 treatment upregulated hepatic proinflammatory factors, such as *Lcn2*, *Tnfa*, *Ppara*, and *Mcp1* (Figure 3E). These results indicated that oral JD5037 treatment increased liver fibrosis, dysregulated BA homeostasis, and induced liver inflammation in *Mdr2^−/−^* mice.

*Long-term JD5037 treatment did not prevent liver fibrosis development*. *Mdr2^−/−^* mice begin developing liver fibrosis at 3 weeks of age. To investigate whether JD5037 treatment prevents the development of liver fibrosis, we orally administered JD5037 to the mice at 3 weeks of age and evaluated its effects at 8 weeks of age (Figure 4A). As depicted in Figure 4B, JD5037 treatment resulted in a statistically insignificant reduction in body weight gain. Sirius red staining revealed exacerbated liver fibrosis in the JD5037-treated group (Figure 4C). Furthermore, the levels of liver injury markers ALT and ALP were elevated by JD5037 treatment (Figure 4D). The serum BA levels were elevated, while liver BA levels remained unchanged (Figure 4E). The prevention study yielded results consistent with the treatment study described earlier.

## 4. Discussion

Previous studies demonstrated that the genetic disruption of Mdr2 (Abcb4), a mouse ortholog of the human MDR3 (ABCB4) gene, caused a depletion of the phosphatidylcholine from bile, leading to liver cholestasis and injury. *Mdr2^−/−^* mice have been extensively used as a genetic model of experimental cholestatic liver injury. In the current study, we used *Mdr2^−/−^* mice to evaluate the therapeutic effect of JD5037, a peripherally restricted CB1R inverse agonist, on cholestatic liver injury.

Endocannabinoids are lipid mediators that function via G-protein-coupled CB1 and CB2 receptors and other molecular targets. The neutral CB1R antagonist rimonabant was effective in reducing body weight [13,14] but with adverse neuropsychiatric effects [15]. In an effort to eliminate these adverse effects, a peripherally restricted CB1R inverse agonist, JD5037, was developed [8]. JD5037 was effective in decreasing body weight, indicating that JD5037 functions via peripheral CB1Rs, rather than having central action [16]. In this study, we showed that in *Mdr2^−/−^* mice, JD5037 decreased body weight gains with unclear underlying mechanisms but that could be a result of reduced food intake or increased energy expenditure, as shown in diet-induced obesity mice [8]. 

JD5037 treatment reduced liver fibrosis in mice treated with CCl_4_ [9]. However, this effect of JD5037 on the reduction in liver fibrosis was not observed in the *Mdr2^−/−^* mice. In fact, JD5037 treatment exacerbated the cholestasis-associated liver injury and fibrosis in the *Mdr2^−/−^* mice. There are two fundamental differences between CCl_4_-induced liver cholestasis and spontaneously developed cholestasis in *Mdr2^−/−^* mice. CCl_4_-induced liver damage is a result of its toxic effects on hepatocytes, while the cholestasis in *Mdr2^−/−^* mice is induced by the disruption of phospholipid transport and bile formation. CCl_4_-induced liver injury may be partially reversible upon the removal of CCl_4_, while *Mdr2^−/−^* induced cholestasis is less likely to be fully reversible. This discrepancy is likely responsible for the different effects of JD5037 treatment. 

In summary, our findings demonstrate that JD5037 treatment exacerbates liver injury and fibrosis in *Mdr2^−/−^* mice. The treatment leads to increased circulating BA levels, enhanced fibrotic deposition, liver ductular reactions, and liver inflammation. These results suggest a detrimental effect of JD5037 in the context of Mdr2 dysfunction or deficiency, cautioning against its use as a therapeutic approach for liver fibrosis associated with cholestasis. Further research is warranted to elucidate the underlying mechanisms and explore alternative therapeutic strategies under this condition.

## Figures and Tables

**Figure 1 cells-13-01101-f001:**
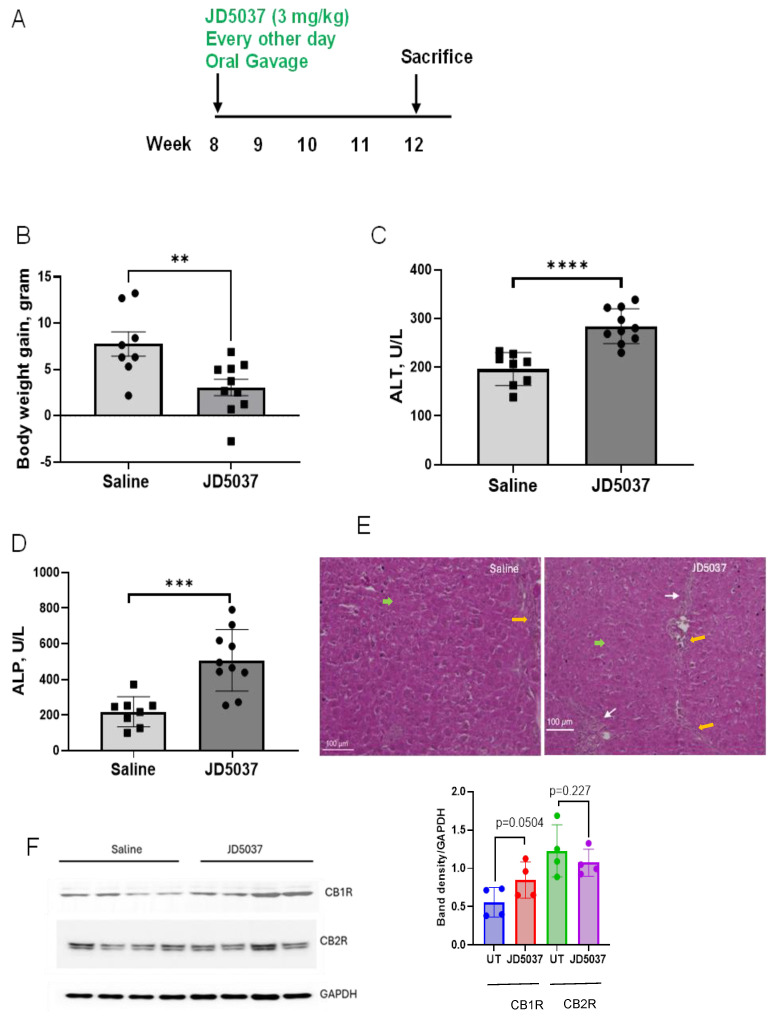
JD5037 treatment caused liver injury in *Mdr2^−/−^* mice. (**A**) Treatment schedule. (**B**) JD5037 reduced body weight. (**C**,**D**) JD5037 increased liver enzymes. (**E**) H&E staining of liver sections. White arrows: periductal fibrosis; yellow arrows: inflammatory cells; green arrows: hepatocyte damage. Magnification, 100×. (**F**) The effects of JD5037 on hepatic CB1R and CB2R protein expression. Left: immunoblots; right: band density analysis, blue dots denote untreated (UT, saline), red denotes JD5037 treated, green denotes untreated, and purple denotes JD5037 treated CB1R or CB2R levels in each animal samples. ** *p* < 0.01; *** *p* < 0.001; **** *p* < 0.0001.

**Figure 2 cells-13-01101-f002:**
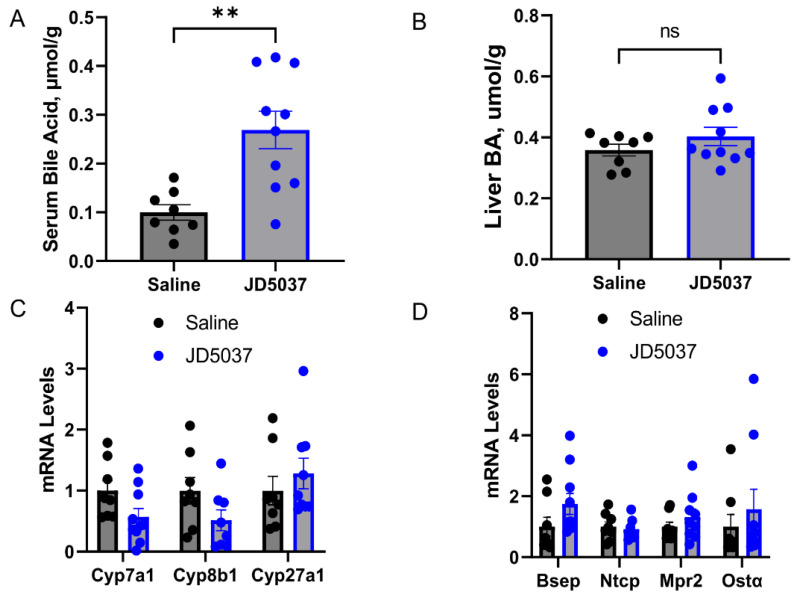
JD5037 treatment increased serum bile acid levels in *Mdr2^−/−^* mice. (**A**) Serum bile acid. (**B**) Liver bile acid. (**C**) mRNA levels of bile acid transporters. (**D**) mRNA levels of the enzymes for bile acid synthesis. Each dot represents one mouse sample. Black dots indicate saline treated, and blue dots denote JD5037 treated. ns denotes non-significant. ** *p* < 0.01.

**Figure 3 cells-13-01101-f003:**
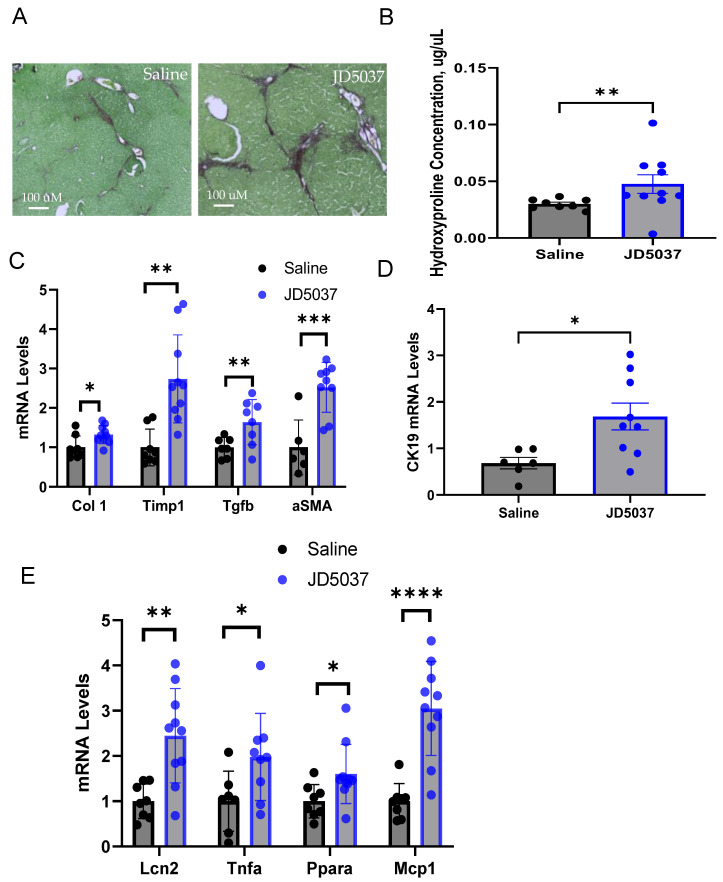
JD5037 treatment exacerbated liver fibrosis in *Mdr2^−/−^* mice. (**A**) Sirius red staining of liver sections. Magnification, 100×. (**B**) Liver hydroxyproline concentrations. (**C**) mRNA levels of fibrogenic genes in the liver. (**D**) CK19 mRNA levels in the liver. (**E**) mRNA levels of inflammatory factors in the liver. Black dotes denote saline treated and blue dots denote JD5037 treated mouse sample. * *p* < 0.05; ** *p* < 0.01; *** *p* < 0.001; **** *p* < 0.0001.

**Figure 4 cells-13-01101-f004:**
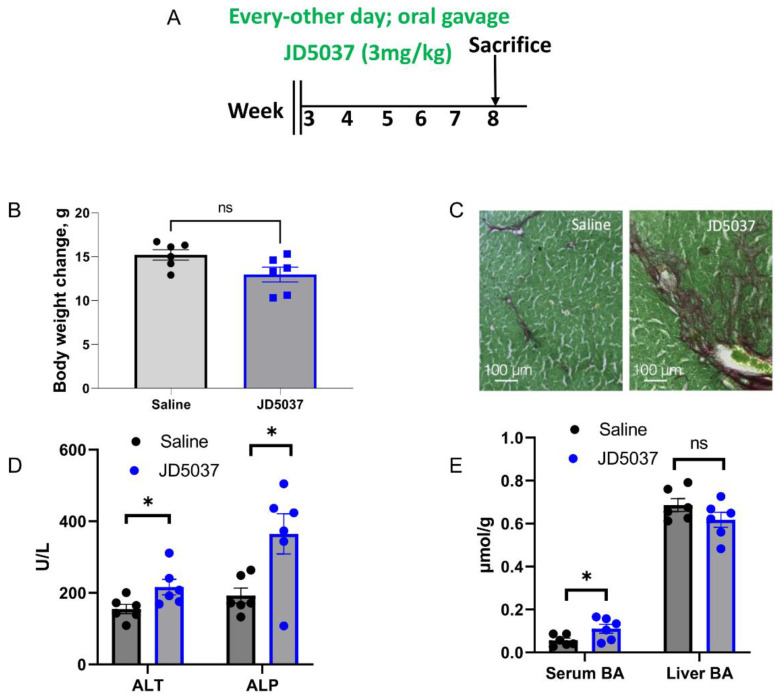
JD5037 treatment did not prevent liber fibrosis development in *Mdr2^−/−^* mice. (**A**) Treatment schedule. (**B**) Body weight gain. (**C**) Sirius red staining of liver sections. Magnification, 100×. (**D**) Liver enzyme levels. (**E**) Bile acid levels in the serum and liver. Black dotes denote saline treated and blue dots denote JD5037 treated mouse sample. ns denotes non-significant. * *p* < 0.05.

**Table 1 cells-13-01101-t001:** Primer sequences for real-time RT-PCR analysis (source: mouse).

Gene	Forward 5′-3′	Reverse 5′-3′	Gene ID	Product Size (bp)
*Lcn2*	TGCCACTCCATCTTTCCTGTT	GGGAGTGCTGGCCAAATAAG	16819	101
*Ppara*	AGAGCCCCATCTGTCCTCTC	ACTGGTAGTCTGCAAAACCAAA	19013	153
*Col1a1*	TCCTCCAGGGATCCAACGA	GGCAGGCGGGAGGTCTT	12842	148
*Tnf*	AGGGTCTGGGCCATAGAACT	CCACCACGCTCTTCTGTCTAC	21926	103
*Ccl2*	GGCCTGCTGTTCACAGTTGC	CCTGCTGCTGGTGATCCTCT	202969	150
*Timp1*	GCATCTCTGGCATCTGGCATC	GGTATAAGGTGGTCTCGTTGA	21857	163
*Acta2*	CTGACAGAGGCACCACTGAA	GAAGGAATAGCCACGCTCAG	11475	288
*Tgfb1*	TTGCTTCAGCTCCACAGAGA	TGGTTGTAGAGGGCAAGGAC	21803	183
*Abcb11*	GCTGCCAAGGATGCTAATGC	CTACCCTTTGCTTCTGCCCA	27413	112
*Slc10a1*	TCATCTGCGGCTGCTCTCC	TGGTCATCACAATGCTGAGGTTC	20493	90
*Slc51a*	TGTTCCAGGTGCTTGTCATCC	CCACTGTTAGCCAAGATGGAGAA	106407	67
*Cyp7a1*	TGGAATAAGGAGAAGGAAAGTA	TGTGTCCAAATGCCTTCGCAGA	13122	269
*Cyp8b1*	CCTCTGGACAAGGGTTTTGTG	GCACCGTGAAGACATCCCC	13124	112
*Cyp27a1*	TGCCTGGGTCGGAGGAT	GAGCCAGGGCAATCTCATACTT	104086	82
*Rn18s-rs5*	GTAACCCGTTGAACCCCATT	CCATCCAATCGGTAGTAGCG	110183	151

## Data Availability

The data presented in this study are available in this article.

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
