# Peer review of "Peripherally Restricted CB1 Receptor Inverse Agonist JD5037 Treatment Exacerbates Liver Injury in MDR2-Deficient Mice"

_cells, 2024, doi:10.3390/cells13131101_

Round 1
Reviewer 1 Report
Comments and Suggestions for Authors
This paper showed the effectiveness of JD3057 in Mdr2-/- mice and the exacerbating effect of JD5037 on liver fibrosis in genetically MDR2-deficient mice. These findings underscore the need for caution in the use of peripherally restricted CB1R inverse agonists for liver fibrosis treatment, particularly in cases of dysfunctional hepatic phospholipid transporter. This paper is negative result. However, this paper result will be useful of basic research of liver fibrosis.
Comments on the Quality of English LanguageI have no claim about the Quality of English language
Author Response
Thank you for your comments. We have done English editing in the revision.
Reviewer #1: This paper showed the effectiveness of JD3057 in Mdr2-/- mice and the exacerbating effect of JD5037 on liver fibrosis in genetically MDR2-deficient mice. These findings underscore the need for caution in the use of peripherally restricted CB1R inverse agonists for liver fibrosis treatment, particularly in cases of dysfunctional hepatic phospholipid transporter. This paper is negative result. However, this paper result will be useful of basic research of liver fibrosis.
Response: We thank you for taking the time in reviewing our manuscript! We appreciate your thoughtful comments.
Reviewer 2 Report
Comments and Suggestions for Authors
In this original manuscript, the authors aimed to investigate the effectiveness of JD3057 in Mdr2 mice. There are several concerns should be addressed before it can be considered for publication.
1. The study design is overly simplistic, lacking any mechanistic investigations.
2. Did the author check the effect of JD5037 in other cholestasis model.
3. How is the distribution of CB1R in Mdr2 liver? Cholangiocytes are the target cells in cholestasis. Does CB1R expression in cholangiocytes?
4. What is the change in the liver of normal mice treated with JD5037? Control groups are requested in this study.
5. Better histological images are requested (unknow black dots in HE images, lower magnification images are requested for Sirius Red). In Figure 3A, the representative Sirius Red images are not match with results (it seems Saline group have more fibrosis).
6. This is a rough manuscript. The manuscript should be extensively revised to meet the format and match results and Figures.
7. IHC for CK19 is requested to evaluate ductular reactions.
8. Most data are present as PCR. Some key data should be confirmed by IHC/IF or WB.
Comments on the Quality of English LanguageN/A
Author Response
Reviewer #2: In this original manuscript, the authors aimed to investigate the effectiveness of JD3057 in Mdr2 mice. There are several concerns should be addressed before it can be considered for publication.
- The study design is overly simplistic, lacking any mechanistic investigations.
Response: First and foremost, we would like to express our gratitude to Reviewer #2 for taking the time to read through and help improve the quality of manuscript. We agree that we could have included more mechanistic investigations and thus write a stronger paper. Unfortunately, the lab was relocated before mechanistic investigations could begin, and the first author could not relocate with the lab to continue with the experiments. Therefore, we decided to submit as a Communication article, which is defined on the Cells site as: “Communications are short articles that present groundbreaking preliminary results or significant findings that are part of a larger study over multiple years. They can also include cutting-edge methods or experiments, and the development of new technology or materials. The structure is similar to an article.” Based on the definition of a Communication article, we feel that our manuscript’s preliminary results meet the criteria, and we will continue to work on mechanistic investigations in our future manuscripts.
- Did the author check the effect of JD5037 in other cholestasis model.
Response: Thank you for your time in reviewing our manuscript. We did cite and check the effects of JD5037 in the cholestasis model induced by CCl4, as described below.
Pg 2, Lines 109-112***, Introduction: “Previous studies have demonstrated that JD5037 reduces body weight gain and attenuates liver fibrosis in mice treated with CCl4. CCl4 is a toxin that induces hepatocellular damage, inflammation, activation of hepatic stellate cells, and subsequent collagen deposition and fibrosis [9]”
Pg 7, Lines 268-270***, Discussion: “Based on previous studies, JD5037 treatment reduces liver fibrosis in mice treated with CCl4 [9]. However, we did not observe a reduction in liver fibrosis when Mdr2-/- mice were treated with JD5037.”
- How is the distribution of CB1R in Mdr2 liver? Cholangiocytes are the target cells in cholestasis. Does CB1R expression in cholangiocytes?
Response: Thank you for your statement. We agree that we should expand our initial introduction into CB1R distribution. We have adjusted our manuscript as such: Pg 2, Lines 91-98***, Introduction:
“In healthy liver tissue, endocannabinoid receptors have minimal expression, but studies have revealed upregulation of CB1R in hepatic myofibroblasts and vascular endothelial cells in various liver diseases, including alcohol-associated liver disease, metabolic dysfunction-associated steatotic liver disease, hepatic fibrogenesis, and hepatic cirrhosis, among others [6]. One study demonstrated that in a chronically damaged liver, CB1R shows its strongest expression in nonparenchymal cells like inflammatory cells, proliferating cholangiocytes, hepatic stellate cells, and liver fibrogenic cells within fibrotic septa [5]”
- What is the change in the liver of normal mice treated with JD5037? Control groups are requested in this study.
Response: Thank you for your insight and request. We agree that the original data would have been more complete with a control group including normal mice treated with JD5037 vs saline, but unfortunately that was never completed with the original experiments. Unfortunately, the lab has now relocated and none of the authors are able to complete any new experiments as a new animal lab has not yet been set up. We feel that the nature of a Communication article can allow for such limitations, and we hope that our current data can guide new experiments and manuscripts for more fleshed out articles in the future.
- Better histological images are requested (unknow black dots in HE images, lower magnification images are requested for Sirius Red). In Figure 3A, the representative Sirius Red images are not match with results (it seems Saline group have more fibrosis).
Response: We appreciate these suggestions and comments. The black dots are unfortunately tiny pellets that were precipitated in our old Hematoxylin dye, and the washing step did not completely wash those away. In Figure 3A, while we can appreciate that both JD5037 and saline treated groups have a large amount of fibrosis due to the nature of the Mdr2-/- genetic model, there are thicker and less organized bands of fibrosis in the JD5037 treated group, as seen when measured on ImageJ. However, we have attached hopefully better images below, with lower magnification images of Sirus Red and additional views of the HE images.
- This is a rough manuscript. The manuscript should be extensively revised to meet the format and match results and Figures.
Response: We sincerely appreciate this input. We have reworked the manuscript based on the Cells provided template and have attached it to this email in hopes that we have now fulfilled this requirement.
- IHC for CK19 is requested to evaluate ductular reactions.
Response: Thank you for you suggestion. CK19 mRNA has been shown in the manuscript in this report. We will perform further characterization, such as IHC, in the future study.
- Most data are present as PCR. Some key data should be confirmed by IHC/IF or WB.
Response: We agree that we could have confirmed/supported key data points with IHC/IF or WB, which would have resulted in a stronger paper. We are continuing to do further research. As a example, we performed a Western blot for CB1R and CB2R in the liver. We shoed that CB1R was increased by JD5037 treatment, while CB1R remained unchanged, as shown in Fig 1F.
Reviewer 3 Report
Comments and Suggestions for Authors
Chen and colleagues investigated the impact of CB1 receptor JD5037 on phospholipid transporter deficiency induced liver fibrosis. The authors used MDR2 deficient mice model, which is characterized by genetic disruption of the Mdr2 gene encoding for the canalicular phospholipid transporter, which resulted in the absence of phosphatidylcholine from bile, leading to liver injury and fibrosis. The study reveals an exacerbating effect Of JD5037 on liver fibrosis in this mouse model as demonstrated by, e.g. intensified liver fibrosis, increased fibrogenic gene expression, stimulated ductular reaction and upregulated hepatic pro-inflammatory cytokines.
The manuscript is well written, however, there are some issues that need to be addressed. In general, please more detailed in the histological analysis. Was the fibrosis evenly distributed or were there focal points? Where exactly do the CB1 receptors occur in liver tissue? Which liver cells have the CB1 receptor?
Material and methods
In table 1, add the product size (bp) and GenBank ID of primer sets.
How is the use of the dose of JD5037 justified?
Results
In Figure 1 Sirius Red staining should be added and in figures 3 and 4 H.E. staining should be included. The images should be magnified to reveal details of the typical morphology of liver tissue, e.g. image sections should show the Glisson trias and sinusoids between hepatocytes in magnification.
In general, Western blots would support the real time RT-PCR results.
Figure 1 and Figure 4 should show similar parameters investigated to compare mouse group (with later start point of treatment and 4 weeks duration of treatment) with that mouse group started at earlier time point of treatment and 5 weeks duration of treatment.
Author Response
Reviewer #3: The manuscript is well written, however, there are some issues that need to be addressed.
- In general, please more detailed in the histological analysis. Was the fibrosis evenly distributed or were there focal points? Where exactly do the CB1 receptors occur in liver tissue? Which liver cells have the CB1 receptor?
Response: We appreciate this feedback, as it is something we had not considered previously. We have adjusted our manuscript as such: Eight-week old mice developed spontaneous fibrosis, as shown there were some focal points, in the liver. Intriguingly, a 4-week treatment with JD5037 dramatically increased evenly distributed liver fibrosis along the liver lobule as indicated by Sirus Red staining of liver sections (Fig 3A). As for the cell type of CB1R expression, we have cited previous studies and revised our introduction as responded to Reviewer 1:
“In healthy liver tissue, endocannabinoid receptors have minimal expression, but studies have revealed upregulation of CB1R in hepatic myofibroblasts and vascular endothelial cells in various liver diseases, including alcohol-associated liver disease, metabolic dysfunction-associated steatotic liver disease, hepatic fibrogenesis, and hepatic cirrhosis, among others [6]. One study demonstrated that in a chronically damaged liver, CB1R shows its strongest expression in nonparenchymal cells like inflammatory cells, proliferating cholangiocytes, hepatic stellate cells, and liver fibrogenic cells within fibrotic septa [5]”
- In table 1, add the product size (bp) and GenBank ID of primer sets.
Response: Thank you for the suggestion. We have included the recommended changes in Table 1 in the revision.
- How is the use of the dose of JD5037 justified?
Response: Previous study on the effect of JD5037 in CCl4-induced liver fibrosis used the dose of 3mg/kg body weight by oral gavage (see reference 9). We used this dose in MDR2 KO mice.
- In Figure 1 Sirius Red staining should be added and in figures 3 and 4 H.E. staining should be included. The images should be magnified to reveal details of the typical morphology of liver tissue, e.g. image sections should show the Glisson trias and sinusoids between hepatocytes in magnification.
Response: We have edited our figures in accordance to this reviewer’s comment, and we thank the reviewer for their insight. We have combined the histological slides from Figures 1 and 3 to demonstrate the HE and Sirus Red stained images. Unfortunately, we did not gather HE images of the prevention group in Figure 4.
- In general, Western blots would support the real time RT-PCR results.
Response: We agree that we could have confirmed/supported key data points with WB, which would have resulted in a stronger paper.
(This could serve as a response to many comments raised by reviewers)
Unfortunately, the lab was relocated after the data presented in the manuscript was completed, and many of the authors could not relocate with the lab to continue with additional experiments. Therefore, we decided to submit our manuscript as a Communication/Repots article, which is defined on the Cells site as: “Communications are short articles that present groundbreaking preliminary results or significant findings that are part of a larger study over multiple years. They can also include cutting-edge methods or experiments, and the development of new technology or materials. The structure is similar to an article.” We feel that our manuscript’s preliminary PCR results meet the criteria of a Communication/Reports article, and we will continue to work on expanding our preliminary data in future experiments, once the lab is set up fully.
- Figure 1 and Figure 4 should show similar parameters investigated to compare mouse group (with later start point of treatment and 4 weeks duration of treatment) with that mouse group started at earlier time point of treatment and 5 weeks duration of treatment.
Response: Thank you for your comments. MD2 KO mice spontaneously developed liver fibrosis. At the age of three weeks, minimal fibrosis can be observed in the liver, while significant liver fibrosis can be observed in week 8. Figure 4 intended only to report that JD5037 has no preventive effects.
Round 2
Reviewer 2 Report
Comments and Suggestions for Authors
The authors have addressed all the comments. I have no more comments.
Author Response
Thank you for you comments
Reviewer 3 Report
Comments and Suggestions for Authors
The authors have made only minor corrections to the reviewer comments.
Table 1 is incomplete (Gene ID o f 18s is missing) and product sizes are not included. The table is not correctly inserted into the text.
Figure 1E. Histological images are not described. Which structures do the arrows point to? Magnifications are of liver tissue as suggested are missing.
Figure 1F. The Western blots are not sufficient and should be supplemented by a diagram quantifying the data as it is usually done.
Author Response
Thank you for your review. Your comments help us to improve the manuscript. We have made the correction/improvement as following:
Table 1 is incomplete (Gene ID o f 18s is missing) and product sizes are not included. The table is not correctly inserted into the text.
Response: Sorry for the missing. We added the Gene ID for 18S in the Table 1. We also added product sizes to the Table. Now the Table is properly inserted into the text. If there is a need for further manuscript formatting, we will follow the editorial instruction to do so.
Figure 1E. Histological images are not described. Which structures do the arrows point to? Magnifications are of liver tissue as suggested are missing.
Response: We have described histological findings in Fig 1E as: “Hematoxylin and Eosin staining of liver sections showed that Mdr2-/- mice at age of 12 weeks had significant cell death and liver architecture disruption. JD5037 administration increased hepatocellular damage (green arrows), inflammatory cell infiltration (yellow arrows) and periductal fibrosis (white arrows) (Fig 1E).” Magnifications were also added.
Figure 1F. The Western blots are not sufficient and should be supplemented by a diagram quantifying the data as it is usually done.
Response: We quantified the Western blot and added in the Fig 1. (right panel of Fig 1F). We added the description in the text: “In addition, hepatic CB1R protein was marginally upregulated by JD5037 treatment, while CB2R protein levels remained unchanged (Fig 1F).
Round 3
Reviewer 3 Report
Comments and Suggestions for Authors
The authors have corrected and improved the paper as suggested by the reviewer. Please carefully check the manuscript according to the guidelines of the journal.
Author Response
Thank you for the suggestion. We have carefully checked the manuscript with the guidelines.